# Fecal Calprotectin Determination in a Cohort of Children with Cow’s Milk Allergy

**DOI:** 10.3390/nu17010194

**Published:** 2025-01-06

**Authors:** Caterina Anania, Filippo Mondì, Giulia Brindisi, Alessandra Spagnoli, Daniela De Canditiis, Arianna Gesmini, Lavinia Marchetti, Alessia Fichera, Maria Grazia Piccioni, Anna Maria Zicari, Francesca Olivero

**Affiliations:** 1Department of Maternal Infantile and Urological Sciences, Sapienza University of Rome, 00161 Rome, Italy; filippo.mondi@uniroma1.it (F.M.); giulia.brindisi@uniroma1.it (G.B.); arianna.gesmini@uniroma1.it (A.G.); lavinia.marchetti@uniroma1.it (L.M.); alessia.fichera@uniroma1.it (A.F.); mariagrazia.piccioni@uniroma1.it (M.G.P.); annamaria.zicari@uniroma1.it (A.M.Z.); 2Department of Public Health and Infectious Diseases, Sapienza University of Rome, 00185 Rome, Italy; alessandra.spagnoli@uniroma1.it; 3Institute of Applied Calculus-CNR Rome, 00185 Rome, Italy; d.decanditiis@iac.cnr.it; 4Independent Researcher, 00100 Rome, Italy; francescaolivero31@gmail.com

**Keywords:** cow’s milk allergy (CMA), non-IgE-mediated food allergy, IgE-mediated food allergy, fecal calprotectin (FC), cow’s milk related symptom score (CoMiSS)

## Abstract

Cow’s milk allergy (CMA) is the most common food allergy among children. An oral food challenge (OFC) remains a mainstay of the diagnosis of CMA, especially for the non-IgE-mediated type; however, this test can be risky and time-consuming. Hence, there is a need to identify biomarkers. Fecal calprotectin (FC) showed variable results, with good reliability and reproducibility in CMA patients. In this prospective study, we enrolled 76 children (aged 5–18 months) with CMA-related gastrointestinal and cutaneous symptoms following guidelines from the European Society of Pediatric Gastroenterology, Hepatology and Nutrition. Clinical assessments included history, physical examination, skin prick tests, and IgE assays. FC levels and the Cow’s Milk Related Symptom Score (CoMiSS) were measured in 51 patients pre (T1) and post-diet (T2), with a subgroup analysis of 15 patients with elevated baseline FC (>50 mg/kg). The results showed that FC levels significantly decreased after the elimination diet (median: 30 mg/kg at T1, 16 mg/kg at T2; *p* < 0.01). In the subgroup with higher FC levels, median values dropped from 90 mg/kg to 33 mg/kg (*p* < 0.01). CoMiSS also improved (median: 8.50 at T1, 3.00 at T2; *p* < 0.01). Linear regression analysis showed no correlation between FC values and the CoMiSS at T1 and T2. In conclusion, the reduction in FC value after an elimination diet suggests that it could be considered a possible biomarker of bowel inflammation in CMA patients. Further studies are necessary to confirm these data and to evaluate and standardize the use of FC for diagnosis and follow-up of CMA.

## 1. Introduction

Food allergy (FA) is defined as an adverse reaction to food due to a specific immune response that reproducibly occurs after the exposure to a given food [1]. Cow’s milk allergy (CMA) is mainly prevalent in infancy [2], with a reported prevalence of 2–7.5% worldwide. In Europe, the prevalence is about 2–3% [3]. In most cases, CMA symptoms occur before one year of age [4], usually during the first weeks of life after the introduction of cow’s milk [5,6]. Then, the prevalence of CMA decreases to less than 1% in children older than six years [6,7]. CMA is caused by a dysregulated immune reaction to one or more milk proteins, and it can be IgE-mediated, non-IgE-mediated, or mixed-type [8]. In some cases, the differentiation between the two forms can be challenging. IgE-mediated CMA occurs as a result of a loss of integrity of key immune components that maintain a state of tolerance [9]. After an initial sensitization and exposure to an allergen (one or more milk’s protein) that is recognized by the immune system as a “pathogen”, there is an activation of the T-helper 2 pathway. Various components are involved in the IgE-mediated response: the epithelium barrier, innate immune cells, T cells, B cells, and effector cells of the allergic response (mast cells, eosinophils, and basophils) [9]. Symptoms of IgE-mediated CMA can affect the skin (rash, urticaria, angioedema, dermatitis), upper and lower respiratory tract (sniffing, wheezing, asthma), and the gastrointestinal tract (abdominal pain, diarrhea, constipation, emesis) [4,9]. Iron deficiency and anemia due to malabsorption represent other manifestations of CMA [10]. Rarely, the neurological and cardio-circulatory systems are involved [9], as in anaphylaxis reactions. Symptoms occur within minutes to one to two hours from the ingestion of the culprit food. In some severe cases, IgE-mediated CMA can manifest as an anaphylaxis reaction. The diagnosis is based on history and clinical examination, and it is confirmed by skin prick tests (SPTs) and determination of specific serum IgE (sIgE) directed against cow’s milk proteins (CMP) [11]. Currently, an oral food challenge (OFC) remains the gold standard to confirm the diagnosis of CMA [1,9]. The pathogenesis of non-IgE-mediated CMA has not yet been fully elucidated. The cell components of the immune system are responsible for non-IgE-mediated CMA; it is not clear to which extent T cells are involved in the cytokine cascade, but their role seems to be significant, especially in food-protein-induced enterocolitis syndrome (FPIES) [12]. Moreover, in non-IgE-mediated CMA, the alteration of the gut epithelial barrier seems to play an important role [13], as well as the modification of the gut microbiota composition. Some studies evaluated the microbiota composition in non-IgE-mediated CMA patients, showing that the genus *Bacteroides* was present in greater numbers in the gut of these patients than in healthy controls [14]. The main symptoms of non-IgE-mediated CMA affect the gastrointestinal system, manifesting as diarrhea, emesis, blood in the stool, irritability, and poor growth. A severe, shock-like reaction can be a manifestation of a non-IgE-mediated food allergy known as FPIES [15]. Non-IgE-mediated CMA also includes food-protein-induced enteropathy (FPE) and food-protein-induced allergic proctocolitis (FPIAP) [16]. In contrast with IgE-mediated CMA, non-IgE-mediated CMA has a delayed onset of symptoms and may have a chronic presentation, making the association with the allergen less evident [17]. The diagnosis of non-IgE-mediated CMA is clinical, except for FPE, where small bowel histology showing villous injury, crypt hyperplasia, and inflammation [15] generally supports the diagnosis. In non-IgE-mediated CMA, SPTs and sIgE are negative; therefore, they are not helpful in the diagnosis, and the gold standard remains the CMP elimination diet for 2–4 weeks, followed by an OFC [4,18]. A careful evaluation of the patient’s clinical history is also essential. The Cow’s Milk-Related Symptom Score (CoMiSS) is a score based on daily duration of crying, frequency, and extent of regurgitation, stool consistency, and skin and respiratory symptoms, utilizing a scale from 0 to 33. It was introduced in 2014 and was updated in 2022 [19]. A score ≥ 10 should enhance awareness of CMA. The CoMiSS score is a useful awareness tool for suggesting and supporting the diagnosis of CMA, as well as in the follow-up of both IgE-mediated and non-IgE-mediated CMA. However, it is not intended as a diagnostic tool and should not replace the OFC. The management of CMA consists of a CMP elimination diet for a variable time period that differs for each child and is guided by individual medical assessment [3,4]. The prognosis of CMA is very good, and children with non-IgE-mediated CMA develop tolerance in a high percentage of cases, usually within three to five years of age. Compared to those with IgE-mediated CMA, the latter usually persists for a longer time [20]. Nevertheless, the individual time to become tolerant is not well known. Guidelines in this regard suggest a follow-up of children affected by CMA every 6 to 12 months, with the reintroduction of cow’s milk after a negative challenge [4]. There is no precise clinical or laboratory criterion that can predict the time of CMP tolerance achievement and, in clinical practice, the OFC can be performed periodically. This often means that a CMP elimination diet is continued for a longer time period than necessary, resulting in discomfort for the patient and the family; furthermore, the nutritional consequences of an unnecessary elimination diet should not be underestimated. The identification of a biomarker that can support the clinician in identifying the achievement of tolerance could be very useful in clinical practice. Several fecal biomarkers have been studied for non-IgE-mediated FA: β-defensin, tumor necrosis factor-α (TNF-α), α-1 antitrypsin (AAT), eosinophil-derived neurotoxin (EDN), eosinophilic cationic protein, and fecal calprotectin (FC) [13]. Calprotectin is a cytosolic protein belonging to the group of S-100 proteins; its fecal concentration increases in conditions such as inflammation, infections, tumors, and in inflammatory bowel disease. It is a calcium- and zinc-binding protein, and therefore has the ability to decrease the local concentration of zinc, depriving some microorganisms of this trace element and inhibiting some zinc-dependent enzymes [21,22]. It has immunomodulatory, antimicrobial, and antiproliferative functions. It is found in the cytoplasm of neutrophils, but also in the membranes of activated macrophages and monocytes, as well as in T and B lymphocytes. FC represents a noninvasive and sensitive marker of gastrointestinal inflammation, as its release in the intestine is related to the passage of neutrophils and mononuclear cells through the intestinal wall, their turnover, and their migration into the intestinal lumen [23,24]. FC concentration correlates with the level of inflammation of the intestinal mucosa, as confirmed by endoscopic and histological investigations in inflammatory bowel disease [25,26]. An increase in FC values has been demonstrated in patients with FA [27]. Previous investigations into the determination of FC in children with CMA have provided variable and inconclusive results on the usefulness of this fecal marker in the diagnosis and follow-up of CMA. Therefore, in this study, we aimed to determine the value of FC in children diagnosed with CMA, aged between 5 and 18 months, before and after a CMP elimination diet, to investigate its possible usefulness in the diagnosis and follow-up of this condition. In addition, we analyzed a possible correlation between FC value and the CoMiSS score.

## 2. Materials and Methods

### 2.1. Study Design

In this prospective study, we enrolled 76 patients diagnosed with CMA (age range 5–18 months), who had been referred to the Pediatric Allergy Unit of the Department of Maternal and Child Health of Sapienza University of Rome between 2020 and 2024, presenting clinical symptoms suggesting CMA, such as digestive complaints (abdominal pain, vomiting, diarrhea, constipation) and skin involvement (rash, urticaria, dermatitis). The diagnosis was established according to the European Society of Pediatric Gastroenterology, Hepatology and Nutrition (ESPGHAN) guidelines and according to a recent position paper (Figure 1) [4,28].

All patients underwent a baseline evaluation with a medical history review and physical examination. To confirm the suspicion of CMA, SPTs along with sIgE to casein, β-lactoglobulin, and α-lactalbumin tests were performed in all patients. In case of anaphylaxis or immediate reactions and positive SPTs, total IgE and sIgE for milk, casein, α-lactalbumin, and β-lactoglobulin tests were performed, and the patients immediately started a CMP elimination diet as a therapy. In the case of gastrointestinal symptoms and negative SPTs, according to the ESPGHAN guidelines [4], we prescribed a diagnostic elimination diet without CMP for 2–4 weeks. In the case of improvement in symptoms, the patient underwent a standardized OFC that, if positive, confirmed a CMP-free diet for a variable time, which was individualized based on the patient. If the challenge was negative, patients were excluded from the study as not being affected by CMA. Determination of FC was performed both before starting the CMA exclusion diet (T1) and after 4 weeks of following the elimination diet (T2) in 51 out of 76 patients. Furthermore, CoMiSS was also recorded at T1 and T2. An additional analysis was performed considering the subgroup of patients that had a more relevant FC basal value at T1 (FC > 50 mg/kg), identifying a total of twenty-two patients; seven of these patients did not complete their follow-up, and fifteen of them had a FC value registered both at T1 and T2. Before fecal samples were collected, infectious diseases, diarrhea, and other confounding factors were ruled out. The exclusion criteria for enrollment were inflammatory bowel disease, celiac disease, diarrhea, vomiting, hematochezia, fever, nasal bleeding in the previous week, consumption of antibiotics, steroidal or non-steroidal anti-inflammatory drugs, and gastric acid inhibitors. A flow diagram of patient recruitment is summarized in Figure 2.

SPTs were performed with CMP; casein, α-lactalbumin, and β-lactoglobulin tests were performed in accordance with the procedure documented by Lieberman et al. [29]. A positive control (histamine) and a negative control (usually saline) for SPTs were always applied. The test was considered positive if the wheal produced by the tested allergen was larger than or equal to 3 mm in diameter compared to the positive control.

We tested serum total IgE and cow’s milk sIgE using a fluorescence enzyme immunoassay (ImmunoCAP, Thermo Fisher Scientific, Uppsala, Sweden).

CoMiSS is a simple, fast, and easy-to-use tool, assessing stool consistency, the daily duration of crying, and the presence and extent of regurgitation, as well as skin and respiratory manifestations. The total score ranges from 0 to a maximum of 33. A cut-off value ≥ 10 was defined by a consensus to identify infants at risk of CMA.

#### 2.1.1. Fecal Calprotectin Measurement

Fecal samples were collected and placed in a sterile stool container (10 mL) without preservatives. The samples were stored in the refrigerator at 4 °C and delivered to the laboratory within 24 h of collection. FC levels were determined using the commercially available enzyme-linked immunosorbent assay (ELISA) kit Calprest^®^, Eurospital’s, Trieste, Italy, according to the manufacturer’s instructions, at enrollment and after the elimination diet. This test involves the use of polyclonal antibodies directed against FC. FC in the diluted sample binds to the antibodies coated on a microplate, constituting a complex with the antibody. Then, the subsequent addition of a chromogenic substrate leads to the formation of a colored mixture. The intensity of the color is proportional to the amount of FC present in the sample. The concentration of FC in the sample is calculated from a 6 standard reference curve. FC levels were measured as mg/kg (µg/g) and interpreted according to the following reference values: <50 mg/kg, negative; 50–100 mg/kg, borderline area (the patients should be retested within a short time), and >100 mg/kg positive. The sensibility of the test is 95%, the specificity is 93%, and the negative predictive value (NPV) is 98%. Given the NPV of this test, false negatives are rare; however, false positives are frequent, because any infectious disease usually increases FC levels.

#### 2.1.2. Ethics

This study was conducted according to the guidelines of the Declaration of Helsinki and approved by the Ethics Committee of the Sapienza University of Rome (project number 5632/2019, prot 858/19).

#### 2.1.3. Informed Consent Statement

Informed oral consent was obtained from all subjects involved in the study.

#### 2.1.4. Statistical Analysis

The data are reported as medians with interquartile ranges (IQR) (25th–75th percentile) for continuous variables and as frequencies and percentages for categorical variables. Since the variables are non-Gaussian, the Wilcoxon rank-sum test was applied to assess differences between two generic groups. Specifically, we tested the differences between FC values at T1 and at T2. This comparison was then repeated by restricting the analysis to patients with FC > 50 mg/kg at T1, and finally, to patients with FC > 50 mg/kg at T1 affected by non-IgE-mediated CMA. Additionally, we tested the differences in CoMiSS values at T1 and T2. All analyses were performed using the software R (version 4.2.2 R Foundation for Statistical Computing, Vienna, Austria) and Matlab R2023a.

## 3. Results

### Patients’ Characteristics

During the study period, FC determination was performed on 51 children: 34 males (66.7%) and 17 females (33.3%) with a mean age of 1.30 years [0.75, 2.20], median weight of 10.50 kg [8.88, 12.75], and median height of 78.00 cm [73.50, 90.50]. Enrolled patients mostly complained of gastrointestinal symptoms, in particular diarrhea/constipation 73%, abdominal pain 41%, and vomiting 44%. The demographic and clinical characteristics of the patients with CMA in which FC determination was performed are summarized in Table 1.

OFC confirmed CMA in all patients. The percentage of IgE and non-IgE-mediated CMA was similar (51% vs. 49%). The median FC value at T1 was 30.00 mg/kg [15.50, 63.90], and at T2 it was 16.00 mg/kg [10.00, 30.00]. The median value of FC is statistically lower at time T2 compared to time T1 (*p* = 0.0014) (Figure 3).

Gender and age did not appear to influence the variation in FC. Among a subgroup of fifteen patients, who had a more relevant FC value at T1 (FC > 50 mg/kg), the reduction in FC from T1 to T2 was statistically significant (*p* value = 0.007), with a median FC at T1 of 90 mg/kg [70, 222.25], which is statistically higher than the median FC value at T2 of 33 mg/kg [21.25, 57.75] (Figure 4).

In this subgroup of fifteen patients, ten were affected by non-IgE-mediated CMA, and five were affected by IgE-mediated CMA. Further restricting the analysis to the ten patients with FC > 50 mg/kg at T1 who had non-IgE-mediated CMA, the difference between the medians of FC at T1 and T2 was statistically significant, with a *p* value of 0.007; specifically, the median FC value at T1 was 103.75 mg/kg [70, 178], while at T2, it was 31.5 mg/kg [18, 50] (Figure 5).

When comparing the two CMA subgroups (IgE- and non-IgE-mediated CMA), most patients who had a significant reduction in their FC value from T1 to T2 were part of the non-IgE-mediated CMA subgroup (Figure 5). However, there was no statistically significant difference between the medians of the FC values of the two subgroups. CoMiSS was performed at T1 and T2 in forty-three of the fifty-one patients in whom FC was measured. The median [IQR] CoMiSS calculated at T1 and T2 were, respectively, 8.50 [6.00, 13.00] and 3.00 [2.00, 5.00]. The median CoMiSS value is lower at T2 compared to time T1 (*p* value < 0.001) (Figure 6).

Gender and age did not influence the variation in CoMiSS either.

Linear regression analysis showed no correlation between FC values and the CoMiSS score either before (T1) or after (T2) the CMP elimination diet (Figure 7).

## 4. Discussion

In the present study, we demonstrate that FC values at diagnosis in children with CMA were significantly higher compared to FC values at follow-up. FC reflects the migration of neutrophils to the intestinal lumen and is considered an accurate biomarker of intestinal inflammation [30]. In addition, FC determination is quickly available given its stability in the stool for more than a week and the possibility to be dosed within a few hours using a simple ELISA test. This test is therefore non-invasive, simple, and relatively inexpensive. FC levels have a central role in gastroenterology, especially in the diagnosis and management of inflammatory bowel disease [31]. FC may also play an important role in FA, and it is hypothesized that in addition to being an inflammatory marker, FC acts as a trigger that amplifies the cascade of inflammatory factors in the allergic response [32]. In FA, eosinophils and neutrophils are activated, while neutrophils and epithelial cells in the intestinal mucosa release calprotectin, resulting in its increase [32]. For these reasons, in the last decade, several studies tried to demonstrate a role for FC determination also in FA, and not only for IBD. In particular, non-IgE-mediated CMA represents a diagnostic challenge since its delayed symptoms may overlap with functional gastrointestinal conditions and, mostly, there is no test to support the diagnosis of this condition. FC determination could prove to be a diagnostic marker of CMA, and it could be useful in the follow-up of this disease. Several published studies showed higher FC values in the CMA group compared to control groups. Baldassare et al. reported higher FC values in 30 infants with non-IgE-mediated CMA than in the control group with values that decreased by 50% after four weeks of an elimination diet [33]. Similar results were confirmed in a prospective study conducted by Trillo Belizon et al. They reported higher FC values in infants aged 1 to 12 months diagnosed with non-IgE-mediated CMA than infants for whom such a diagnosis had been excluded and healthy controls. Moreover, they reported a statistically progressive decline in FC values after one to three months of a CMP elimination diet, with significant differences in patients with gastrointestinal symptoms such as diarrhea or rectal bleeding [34]. The authors concluded that an FC value of 138 mg/kg is a useful cut-off to exclude the diagnosis of non-IgE-mediated CMA. Beser et al. conducted a study including 32 infants under two years of age, who were diagnosed with IgE-mediated and non-IgE-mediated CMA by OFC. FC values were higher in these patients before starting a CMP elimination diet in both IgE-mediated and non-IgE-mediated CMA patients, and this was statistically significant (*p* < 0.001). In addition, FC values before following the CMP elimination diet were higher than in the control group. The authors suggested a possible use of FC in the diagnosis and monitoring of gastrointestinal forms of both IgE-mediated and non-IgE mediated CMA [35]. These studies demonstrate a possible use of FC as a biomarker for the diagnosis and follow-up of CMA, especially in non-IgE-mediated CMA. In a recent systematic review and meta-analysis that included 12 studies, Zhang et al. concluded that FC could be a simple and reliable biomarker for diagnosing CMA, especially non-IgE-mediated CMA in infants less than 12 months; FC was also described as a potential marker for therapy responses [36]. However, they concluded that other factors, such as age and geographical origin, could influence the FC determination. Also, in a prospective case–control trial, Qiu et al. compared 90 CMP allergic and 90 non-allergic infants and found that the median FC value was significantly higher in the allergic group. In addition, FC values in infants with CMA were significantly lower after one month of dietary intervention and decreased further after two months [37]. However, other studies reported conflicting results to those mentioned above. Some studies found no statistically significant differences between FC values at the time of CMA diagnosis and after a diet without CMP, or in comparison with healthy controls. A review conducted by Xiong et al., examining 13 studies with different study designs, concluded that the available evidence is not sufficient to confirm the utilization of FC both in the diagnosis and monitoring of CMA, as well as for predicting allergic disease [27]. Similar results were found by Ambroszkiewicz et al. [8] and Roca et al. [38] in their studies. In a recent paper, ESPGHAN recommends against the use of FC either as diagnostic tool or as a prognostic marker of CMA in children, because current studies show conflicting results [39]. Merras-Salmio et al. collected stool samples during a CMP-free diet and after a challenge and found higher FC values in patients on an elimination diet with a positive challenge than in patients with a negative challenge to CMP or in the controls, demonstrating the presence of mild inflammation of the intestinal mucosa during the challenge [40]. Recently, Lendvai-Emmert et al. conducted a longitudinal cohort study investigating 47 children aged 1 to 18 months who were affected by CMA. The authors found a significant decrease in FC among patients who adhered to a strict elimination diet, and they concluded that FC can be an objective marker in confirming the diagnosis of CMA [41]. A recent study by Zain-Alabedeen et al. investigated FC determination in 120 children, 60 with positive and 60 with negative CoMiSS scores. FC values were higher in the first group (*p* < 0.0001). Unfortunately, one limitation of this study is that the population was not subgrouped into IgE and non-IgE-mediated patients [42]. Our study shows, in accordance with the findings of Beser et al. [35], a statistically significant decrease in FC after an elimination diet in children with CMA *(p* < 0.001). Patient sex and age do not appear to influence FC variation and the percentage of IgE and non-IgE-mediated CMA is similar (51% vs. 49%). It could be arguable that patients with a baseline relevant value of FC at T1 should be the only ones taken into consideration for the purpose of the study. Furthermore, the presence of enrolled patients under one year of age could also be problematic. In fact, although being a reliable biomarker of intestinal inflammation, FC does not have a validated cut-off in smaller children, particularly in infants younger than one year of age. In this population, FC may be elevated without an underlying cause of inflammation and should be cautiously interpreted in the clinical context, as an established cut-off for this age is lacking. As for older children, the 2020 ESPGHAN position paper on the use of FC testing suggests that in children older than 4 years of age cut-off values of 50 mg/kg, as in adults, can be used, although healthy children may have FC values up to 100 mg/kg or even higher [38]. This is why an additional subgroup analysis of the patients who had a T1 FC > 50 mg/kg was performed, confirming the significant reduction in FC at T2. Moreover, in the study by Lendvai-Emmert, no significant difference in FC values was observed in children both under and older than 4 years [41]. Another important limitation is the small sample size of subjects enrolled in the study and the conspicuous number of patients that were lost at follow-up. A larger population would be ideal to understand the real role of FC in the diagnosis of CMA in children. Concerning our secondary endpoint, no correlation between FC value and CoMiSS score was found. In consideration of these results, which are in accordance with other studies mentioned above, we can therefore state that FC determination in CMA patients may be helpful for both diagnosis and follow-up of this condition. Certainly, the aforementioned limitations of the study must be considered.

## 5. Conclusions

In our study, we found that FC values at diagnosis in children affected by CMA were significantly higher compared to FC values after a CMP elimination diet, confirming the data found in the literature. The present study demonstrates the possible usefulness of determining FC at diagnosis and during the follow-up of CMA in children and represents a further contribution to the evidence in this field. However, more well-conducted studies, possibly with a larger sample size, are needed to consolidate these findings and standardize the use of FC for the diagnosis and follow-up of CMA.

## Figures and Tables

**Figure 1 nutrients-17-00194-f001:**
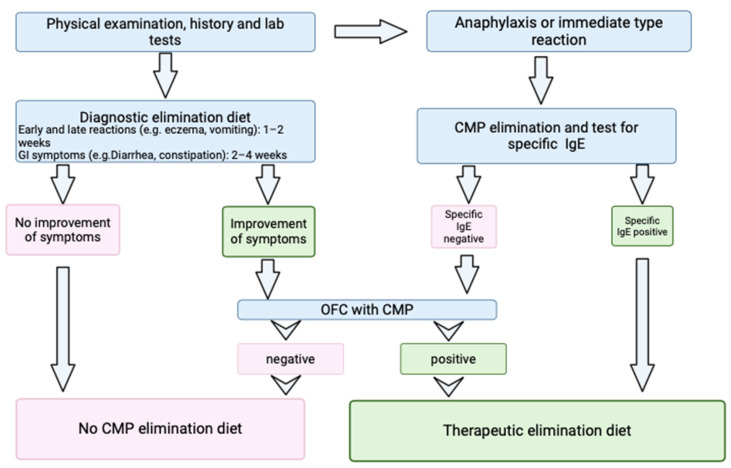
Diagnostic algorithm for cow’s milk allergy in children. (GI: gastrointestinal, OFC: oral food challenge, CMP: cow’s milk proteins).

**Figure 2 nutrients-17-00194-f002:**
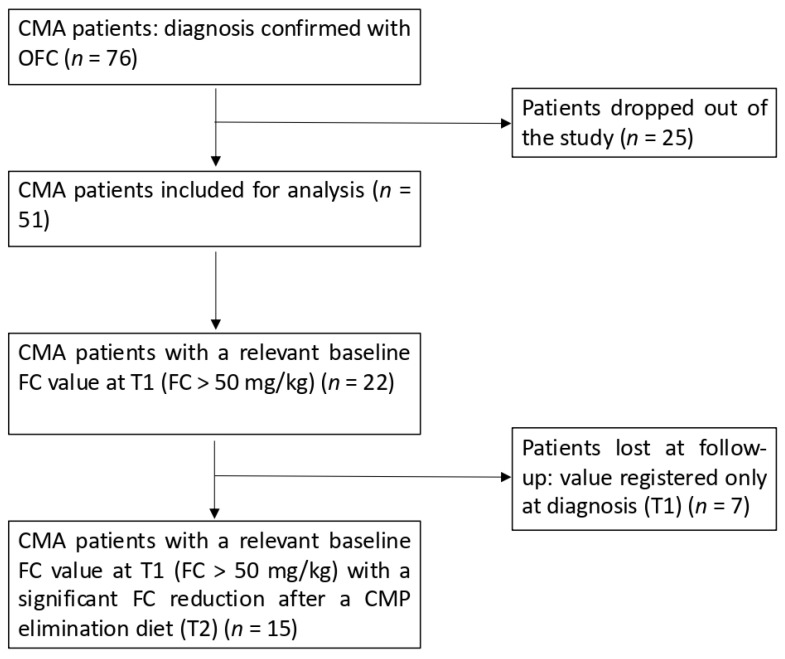
Flowchart of the study.

**Figure 3 nutrients-17-00194-f003:**
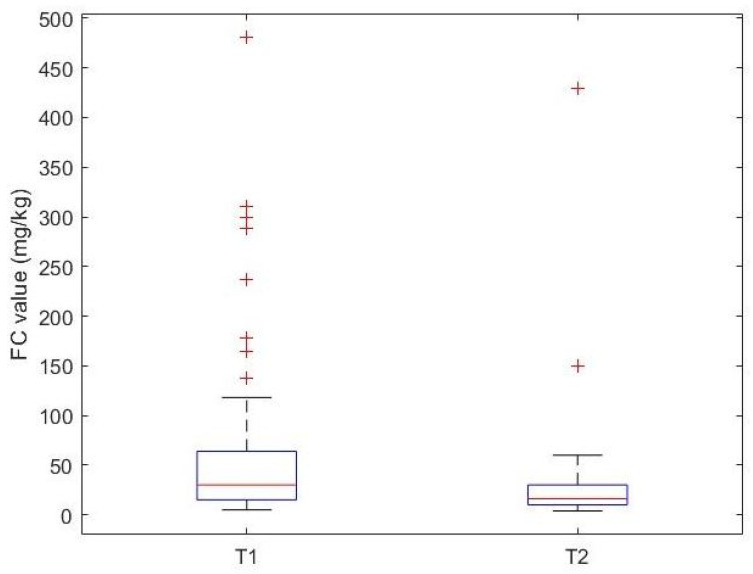
Box plot showing FC values before (T1) and after (T2) the elimination diet in CMA patients.

**Figure 4 nutrients-17-00194-f004:**
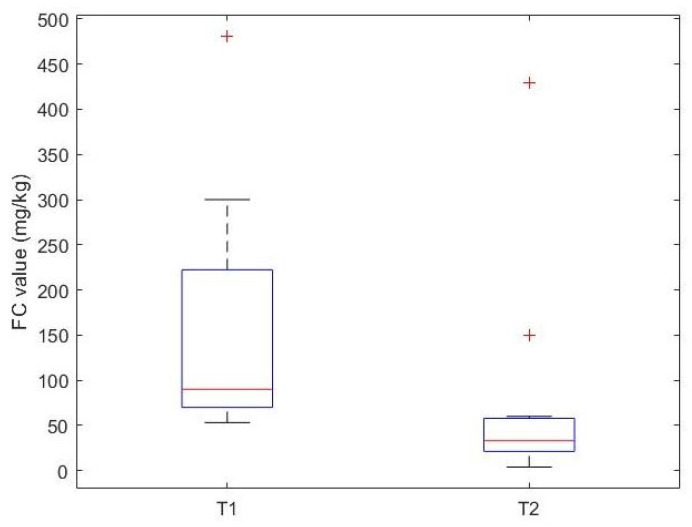
Box plot of FC levels before (T1) and after (T2) elimination diet in a subgroup of CMA children with elevated FC levels before the elimination diet (T1 FC > 50 mg/kg).

**Figure 5 nutrients-17-00194-f005:**
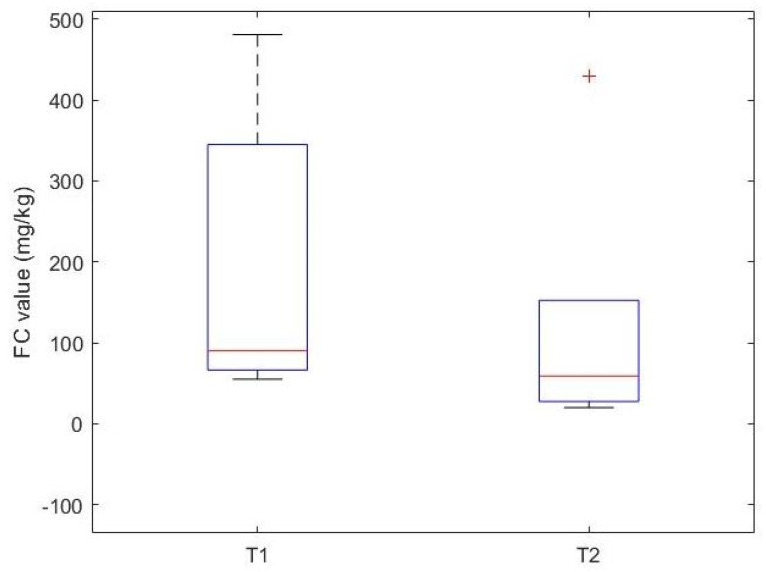
Boxplot showing FC values in a subgroup of children affected by non-IgE-mediated CMA, with elevated FC at T1 (T1 FC > 50 mg/kg) before and after the elimination diet.

**Figure 6 nutrients-17-00194-f006:**
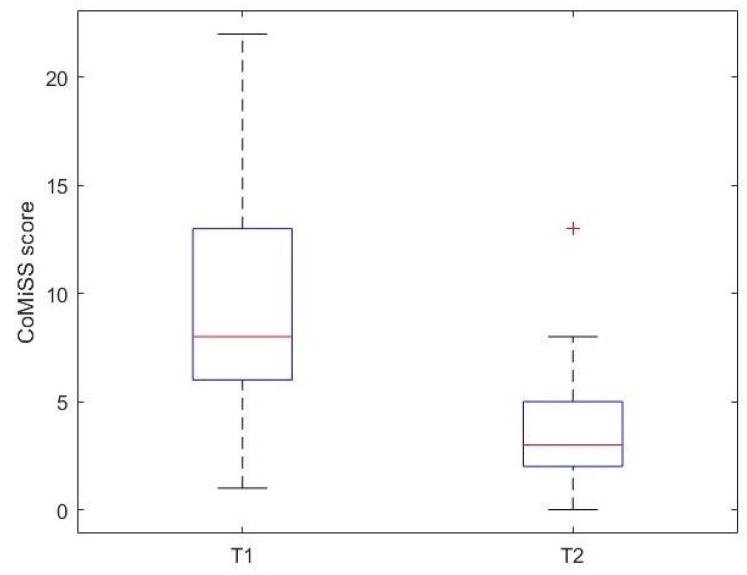
Boxplot of the comparison between CoMiSS score before (T1) and after (T2) the elimination diet in CMA children.

**Figure 7 nutrients-17-00194-f007:**
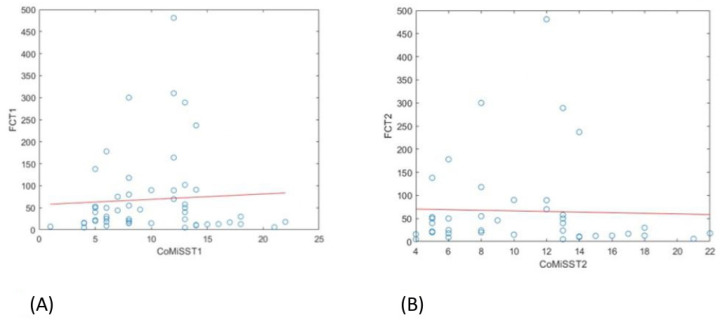
(**A**) Linear regression analysis showing the relationship between FC values and CoMiSS scores before starting the CMP elimination diet. (**B**) Linear regression analysis showing the relationship between FC values and CoMiSS scores after CMP elimination diet.

**Table 1 nutrients-17-00194-t001:** Characteristics of the study population.

Patients’ Characteristics	(*n* = 51)
Boys/girls	34/17 (67/33%)
Age (years)	1.3 [0.7, 2.2]
Weight (kg)	10.5 [8.8, 12.8]
Height (cm)	78.0 [73.5, 90.5]
Breastfed/formula fed/mixed feeding	25/15/11 (49/30/21%)
IgE-mediated/non-IgE-mediated CMA	25/26 (49/51%)
Tot IgE > 15/<15 (kU/L)	22/28 (56/44%)
sIgE to CM > 0.35/<0.35 kU/L	29/21 (58/42%)
sIgE to α-lactalbumin > 0.35/<0.35 kU/L	13/37 (74/26%)
sIgE to β-lactoglobulin > 0.35/<0.35 kU/L	18/55 (25/75%)
sIgE to casein > 0.35/<0.35 kU/L	14/36 (28/72%)
Symptoms	
Vomiting	20 (44%)
Diarrhea/Constipation	37 (73%)
Abdominal pain	21 (41%)
Skin involvement (urticaria, dermatitis)	38 (75%)
Measurements	
Calprotectin T1 (mg/kg)	30 [15.5, 63.9]
Calprotectin T2 (mg/kg)	16.1 [10.0, 30.0]
CoMiSS T1	8.5 [6.0, 13.0]
CoMiSS T2	3.0 [2.0, 5.0]

*n*: number of patients included for analysis, CMA: cow’s milk allergy, CM: cow’s milk, Tot IgE: total serum immunoglobulin E, sIgE: specific serum IgE, CoMiSS: Cow’s Milk-related Symptom Score, T1: before elimination diet, T2: after elimination diet.

## Data Availability

The data used in this study are available from the corresponding. Author upon reasonable request. The data are not publicly available due to privacy reasons.

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
