# Peer review of "Fecal Calprotectin Determination in a Cohort of Children with Cow’s Milk Allergy"

_nutrients, 2025, doi:10.3390/nu17010194_

Round 1
Reviewer 1 Report
Comments and Suggestions for Authors
Reviewer’s Comments and Suggestions for Authors
Journal: Nutrients, MDPI
Manuscript ID: nutrients-3361805
Type: Article
Title: FECAL CALPROTECTIN DETERMINATION IN A COHORT OF CHILDREN WITH COW’S MILK ALLERGY
Authors: Caterina Anania, Filippo Mondì, Giulia Brindisi, Alessandra Spagnoli, Daniela De Canditiis, Arianna Gesmini, Lavinia Marchetti, Alessia Fichera, Maria Grazia Piccioni, Anna Maria Zicari and Francesca Olivero.
The authors of the Manuscript ID: nutrients-3134834 enrolled 76 children (age 5-18 months) with cow’s milk allergy (CMA)-related gastrointestinal and cutaneous symptoms in a prospective study. Fecal calprotectin (FC) levels and Cow’s Milk Related Symptom Score (CoMiSS) were measured in 51 patients pre- (T1) and post-diet (T2), with a subgroup analysis of 15 patients with elevated baseline FC (>50 mg/kg). The authors reported that FC levels significantly decreased after the elimination diet (median: 30 mg/kg at T1, 16 mg/kg at T2; p < 0.01). In the subgroup with higher FC levels, median values dropped from 90 mg/kg to 33 mg/kg (p < 0.01). CoMiSS also improved (median: 8.50 at T1, 3.00 at T2; p <0.01). Linear regression analysis showed no correlation between FC values and the CoMiSS at T1 and T2. The authors stated that the reduction of FC value could be considered a possible biomarker of bowel inflammation in CMA patients.
The authors have described the main limitations of this study. This manuscript should be extensively revised, and the essential revisions should be completed as below.
Major revisions
1. In the title: Please write the Title in lowercase letters, except the first letter of each word.
2. In the abstract: Please remove the words of “Background:”, and “Methods:”, and change the “Results:” to “The resulted showed that...”, and the “Conclusions:” to “In conclusion, ...”.
3. In the Materials and Methods section, certain essential information regarding the experiments of this study was missing. For example,
(1) How did the authors perform the sIgE for casein, beta-lacto globulin and alpha-albumin assays? Please state the name of the manufacturer, city, and country from where the equipment and/or kits used in this study were sourced.
(2) How much fecal sample from each patient was subjected for the FC level assay?
(3) Did the authors perform the assays in this study in triplicate?
(4) Fig. 1.: Please change the “(GI=gastrointestinal” to “GI: gastrointestinal;”. The same issue for the other abbreviations.
4. In the Results section, for example:
(5) Table 1: This table should be reformatted. What did the “Age” mean “1,3”? What did the “Weight (kg)- [median]” mean “[8,8,12.8]”? The same issues for the “Height (cm) [median]”, “Calprotectin (mg/kg) (median, [IQR])”, and “ComiSS”. Please clarify. Abbreviations should be written out in full as a note below the table.
(6) Figure 2: What did the X and Y axes mean, respectively? What did the “+” mean? The authors show mark p values showing statistical difference in FC levels before (T1) and after (T2) the elimination diet in CMA children. The same issues in the other figures. Please clarify.
(7) Figure 7 and Figure 8 should be combined into one figure, as A, and B.
5. In the discussion section, for example:
(8) Lines 286-288: Please rephrase the sentence.
Minor revisions
(9) Please change the “n=76” to “n = 76”, “ (N=51)” to “(n = 51), “p <0.01” to “p < 0.01”, “(p-value=0.0014)” to “p = 0.0014”, “Fig 6” to “Fig. 6”, etc. Please amend the similar issues throughout the article.
(10) Line 263: please remove the “Please add:”.
(11) Please format the References, and carefully check each of the references.
(12) A number of English or typing errors are present in the manuscript. The authors should extensively check and carefully revise throughout the manuscript.
Comments on the Quality of English Language
Moderate editing of English language required.
Author Response
Thank you very much for taking the time to review this manuscript. Please find the detailed responses below and the corresponding revisions/corrections highlighted/in track changes in the re-submitted files:
Major revisions
Point 1: In the title. Please write the Title in lowercase letters, except the first letter of each word.
Response 1: We made this change in the text.
Point 2: In the abstract: Please remove the words of “Background:”, and “Methods:”, and change the “Results:” to “The resulted showed that…”, and the “Conclusions:” to “In conclusion,….”
Response 2: We made this, following the Authors’ guidelines, however we have deleted
these sections in the abstract and made the corrections suggested.
Point 3: In the Materials and Methods section, certain essential information regarding the experiments of this study was missing. For example,
- How did the authors perform the sIgE for casein, beta-lacto globulin and alpha-albumin assays? Please state the name of the manufacturer, city, and country from where the equipment and/or kits used in this study were sourced.
- How much fecal sample from each patient was subjected for the FC level assay?
- Did the authors perform the assays in this study in triplicate?
- 1: Please change the “(GI=gastrointestinal” to “GI:gastrointestinal;”. The same issue for the other abbreviations.
Response 3:
(1) We already provided this information in the text, see lines 175-176” We tested serum total IgE and cow's milk sIgE using a fluorescence enzyme immunoassay (ImmunoCAP, Thermo Fisher Scientific, Uppsala, Sweden) “.
(2) We collected the samples in a standard stool container (10 ml). See line 183
(3) No, the exam is done in a single evaluation as per the protocol; triplicate copies are not required
(4) We have changed “(GI=gastrointestinal” to “GI:gastrointestinal;” and the other abbreviations line 142
Point 4: In the Results section, for example:
- Table 1: This table should be reformatted. What did the “Age” mean “1,3”? What did the “Weight (kg)-[median] mean “[8,8,12.8]? The same issues for the “Height (cm) [median], “Calprotectin (mg/kg) (median, [IQR])”, and “ComiSS”. Please clarify. Abbreviations should be written out in full as a note below the table.
- Figure 2: What did the X and Y axes mean, respectively? What did the “+” mean? The
authors show mark p values showing statistical difference in FC levels before (T1) and after (T2) the elimination diet in CMA children. The same issues in the other figures. Please clarify.
- Figure 7 and Figure 8 should be combined into one figure, as A, and B.
Response 4:
- We have reformatted Table 1 and we have written the abbreviations out in full as a note below the table. Lines 227-246
- We added the indication in the X and Y axes in all the figures
- The figure resulting from the combination of figures 6 and 7 results in lower magnification, which is why we thought it would be better to keep the two figures separate. Anyway, we combined the two figures, as suggested.
Point 5. In the discussion section, for example:
- Lines 286-288: Please rephrase the sentence.
Response 5: We rephrase the sentence. Lines 388-393
Minor Revisions
Point 9: Please change the “n=76” to “n = 76”, “ (N=51)” to “(n = 51), “p <0.01” to “p < 0.01”, “(p-value=0.0014)” to “p = 0.0014”, “Fig 6” to “Fig. 6”, etc. Please amend the similar issues throughout the article.
Response 9: We provided these corrections in the text.
Point 10: Line 263: please remove the “Please add:”.
Response 10: we have removed “Please add”. Line 472
Point 11: Please format the References, and carefully check each of the references.
Response 11: We have checked the references following Authors’s guidelines.
Point 12: A number of English or typing errors are present in the manuscript. The authors should extensively check and carefully revise throughout the manuscript.
Response 12: We revised and checked the manuscript by correcting typing and English errors as suggested.
Thank you for your comments and suggestions.
Reviewer 2 Report
Comments and Suggestions for Authors
The authors have examined calprotectin in case of cow's milk allergy. Relevant findings, supporting previous results.
1.) Do you have information about breastfeeding history of patients?
2.) Is there a difference between breastfed and formulafed infants test results?
3.) How specific fecal calprotectin is in the detection of CMA?
Comments on the Quality of English Language
Fine.
Author Response
Thank you very much for taking the time to review this manuscript. Please find the detailed responses below and the corresponding revisions/corrections highlighted/in track changes in the re-submitted files:
Point 1: Do you have information about breastfeeding history of patients?
Response 1: We added this information in table 1, as you suggested,
Point 2: Is there a difference between breastfed and formulafed infants test results?
Response 2: Unfortunately, we did not evaluate this correlation
Point 3: How specific fecal calprotectin is in the detection of CMA?
Response 3: We do not have conclusive information about this topic.
Thank you for your comments and suggestions.
Round 2
Reviewer 1 Report
Comments and Suggestions for Authors
Reviewer’s Comments and Suggestions for Authors
(Second round)
Journal: Nutrients, MDPI
Manuscript ID: nutrients-3361805
Type: Article
Title: Fecal Calprotectin Determination in a Cohort of Children with Cow’s Milk Allergy
Authors: Caterina Anania, Filippo Mondì, Giulia Brindisi, Alessandra Spagnoli, Daniela De Canditiis, Arianna Gesmini, Lavinia Marchetti, Alessia Fichera, Maria Grazia Piccioni, Anna Maria Zicari and Francesca Olivero.
The authors have revised the manuscript Manuscript ID: biomedicines-2965165, according to most of my comments and suggestions. Revisions should still be done, for example:
(1) Line 27: “In conclusion,”
(2) Table 1: This table is still presented in a low quality. What did the second column (n = 51) mean? Please clarify. The authors should further format and revise many typing and or expression errors of this table.
(3) Figures 7: Please rephrase the figure legend.
Comments on the Quality of English Language
Minor editing of English Language required.
